# Postpartum versus postnatal period: Do the name and duration matter?

**Malith Kumarasinghe**, **Manoja P. Herath, Andrew P. Hills, Kiran D. K. Ahuja** *

School of Health Sciences, University of Tasmania, Launceston, Tasmania, Australia

* Kiran.Ahuja@utas.edu.au

## Abstract

### Introduction

Guidelines and other strategic documents were collated to understand the extent of the global use of terms postpartum and postnatal along with the duration and schedule of maternal care *after* delivery.

### Methods

Postpartum care guidelines and strategies published in English, by international organisations including the World Health Organization, and countries in either the Organization for Economic Co-operation and Development or Group of 20 were included in this scoping review. All documents available online with unrestricted access and published before May 31, 2023, were included. The evolution of the World Health Organization's definition of the period *after* delivery for mothers and the changes in the schedule of routine maternal care following delivery over time were displayed pictorially. A summary table was then developed to present the level of similarities and differences in the latest available documents from the international organisations and countries belonging to either the Organisation for Economic Co-operation and Development or the Group of 20.

### Results

Ten documents from the World Health Organization, one from the European Board, and 15 country-level guidelines from six countries met the inclusion criteria. The interchangeable use of 'postpartum' and 'postnatal' is common. While the World Health Organization mentions the definitive length (six weeks) of the postpartum/ postnatal period, it is not stated in documents from other organisations and countries. Additionally, the length and schedule of routine maternal care *after* delivery vary substantially between organisations/countries, spanning from six weeks to one year with two to six healthcare contacts, respectively.

### Conclusion

Through this review, we make a case for a universal harmonisation of the term postpartum when referring to mothers *after* delivery; add clarity to the documents on the rationale for

**Data Availability Statement:** All relevant data are within the manuscript and its Supporting Information files.

**Funding:** Malith Kumarasinghe was supported by a Tasmania Graduate Research Scholarship Stipend and RTP Fee-Offset Scholarship.

**Competing interests:** The authors have declared that no competing interests exist.

and duration of the postpartum period; and extend the routine maternal care schedule *after* delivery to support women in this vulnerable period.

## Introduction

'Postpartum' and 'postnatal' are commonly used terms when referring to the mother *after* delivery [1, 2]. Various organisations and countries prefer different terminologies, and some use the two terms interchangeably, causing confusion [1, 2]. The ambiguity in using different terms is further exaggerated when considering the length of the period after delivery i.e., the postpartum/postnatal period [3, 4].

Physiologically, almost one year is required for most organs and systems to return to the pre-pregnancy stage [5]. The period after delivery is commonly divided into three phases: immediate, early, and late postpartum phase [2, 4, 6]. While most agree that the immediate phase commences soon after delivery, there is no consensus on the duration of each phase and when the postpartum/postnatal period concludes. For example, some indicate the immediate phase as the first 24 hours [2], whereas others suggest 6–12 hours post-delivery [4, 6]. The early phase usually commences between 12–24 hours [2, 4, 6] and concludes between the seventh day and sixth week, after delivery [2, 4, 6]. The late phase commences between the eighth day and the sixth week and concludes between the sixth week and the sixth month after delivery [2, 4, 6].

Routine postpartum care is a standard package of services provided to mothers following delivery worldwide. The components of the care package change with the phases of the postpartum period. Overall, the package includes care for the mother immediately after delivery, followed by general well-being and any pregnancy-related issues, psychological screening, and support, breastfeeding, and contraception throughout the postpartum period [3]. There is also variability in the recommended schedule (duration and timing) of postpartum care; while most countries follow routine care packages for 42 days (six weeks) for the mother *after* delivery [7–9], some recommend different schedules lasting up to one year [3, 10–12].

Considering this diversity, we first aimed to collate historical guidelines published by the WHO, then describe the differences in the latest guidelines from leading organisations and countries belonging to the Group of 20 (G20) and/or Organization for Economic Co-operation and Development (OECD). This approach was used to understand the progression (if any) in the terminology, duration, and schedule of routine services for the assumed postpartum period. We anticipate that this review will stimulate discussion in the scientific and clinical community to consider harmonising the terminology, length of the postpartum period, and its related provision of routine services worldwide to better support the health of mothers through this vulnerable period.

## Materials and methods

Guidelines and strategy documents on routine postpartum care, published in English, available online for unrestricted access, and published before 31 May 2023 were included in this scoping review. The documents were issued by one of the following organisations/ countries:

- WHO and its regional offices

- European Board & College of Obstetricians and Gynaecologists (EBCOG)

- United Nations Children's Fund (UNICEF)

- International Federation of Gynaecology and Obstetrics (FIGO)

- Nordic Federation of Societies of Obstetrics and Gynaecology (NFOG)

- Authorised institutions of all countries belonging to either G20 or OECD

Guidelines restricted to a specific disease or subtopic related to the period after delivery, such as postpartum haemorrhage or gestational diabetes, were excluded.

While WHO guidelines are widely used, including by many poor-resourced developing and under-developed countries, the countries in G20 and OECD countries are distributed across all continents and have large populations with diverse cultural and socio-economic status. Hence, the eligibility criteria outlined allowed for the collection of relevant, up-to-date global information.

A multipronged search strategy was used to trace the online publication section of organisations (such as FIGO, UNICEF etc) with an individualised search strategy. For a country-specific guidelines, we used a two-step search strategy. We first searched the fifth round of the Global Sexual, Reproductive, Maternal, Newborn, Child and Adolescent Health Policy Survey in the WHO repository [13]; and then conducted an online search of official webpages of the responsible authority of the individual countries e.g., Health Department or Ministry of Health.

All retrieved documents were imported to EndNote20 before removing duplicates. The summary, introduction, and conclusions of all documents were screened by the first author (MK). In the case of uncertain eligibility, a second investigator evaluated the document, and a consensus was reached. Reference lists of the selected guidelines and strategy documents were also screened to identify any other relevant documents. Reasons for the exclusion of any reports were noted and relevant data was extracted.

The evolution of the WHO's definition of the period after delivery for mothers and the changes in the schedule of routine maternal care following delivery over time were displayed pictorially. A summary table was developed to present the latest available information on the definition, duration, and healthcare schedule utilised across organizations, G20, and OECD countries.

Ethics approval was not required for this scoping review.

## Results

The initial search yielded 47 guideline/strategy documents from three international organisations and 57 documents from 28 countries. A bibliographic search yielded a further 8 documents by international organisations and a single country-level document. Therefore, the total number of records identified was 55 from five international organisations and 58 from 28 countries (Fig 1). The final count of eligible documents after excluding those in a language other than English, or documents on specific diseases, was 26. This included 10 documents from the WHO, one from the European Board, and 15 country-level guidelines from six countries: namely Australia, Canada, India, South Africa, the United Kingdom (UK), and the United States of America (USA) (Fig 1, S1 File).

The most recent guidelines concerning routine maternal care aspects after delivery released by international organisations were published between 2014 (EBCOG) and 2022 (WHO), whereas the country-level guidelines (Table 1) were published between 2016 (South Africa) and 2021 (India and the UK). The largest number of documents in English relevant to routine postpartum aspects were from India (n = 5; from 2009 to 2021), followed by South Africa (n = 4, from 2016 to 2023).

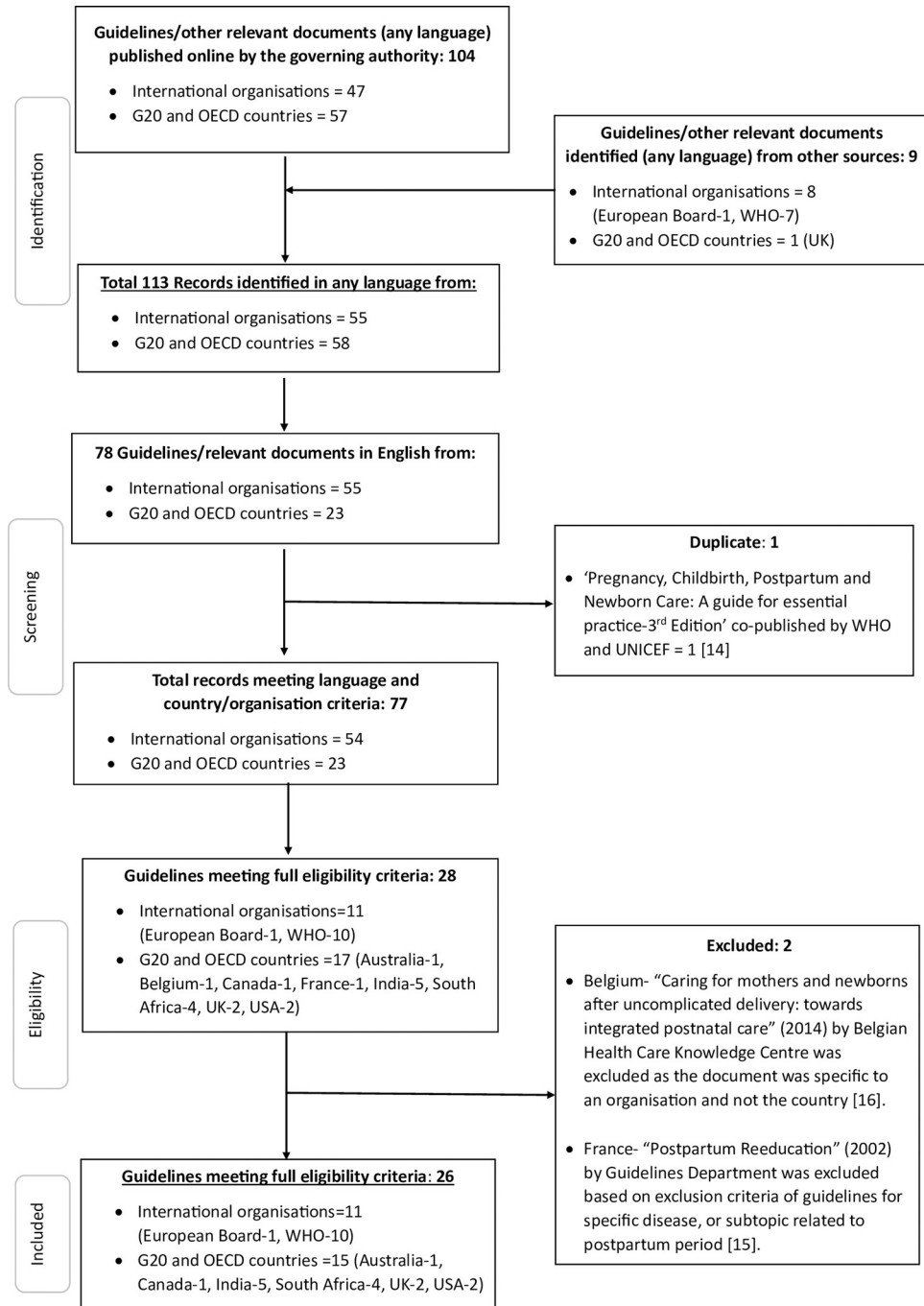

**Fig 1. Progress of guideline and strategy documents from identification to inclusion to review of the terminology associated with the postpartum period and schedule and length of services integrated into routine care.** [14–16].

## WHO guidelines: Evolution of terminology concerning the postpartum period and provision of routine care over time

A review of 10 documents (1976, 1996, 1998, 2003, 2009, 2010, 2013 –two documents, 2015 and 2022) illustrated that both 'postpartum' and 'postnatal' terms have been used interchangeably to refer to the mother following childbirth. The only exceptions were the 1976 and 1996

**Table 1. Definition, duration, length, and frequency of services in routine maternal care *after* delivery included in the latest guidelines from international organisations, and G20 and OECD countries[a].**

| International Body/ Country Institution | Year of publication of the latest guideline/ Position Paper/ Strategy | Use of 'Postpartum' or 'Postnatal' term to refer to care/ services of mothers *after* delivery* | | Postpartum/ postnatal period | Postpartum/ postnatal care provision | |
|---|---|---|---|---|---|---|
| | | Mother | | Duration and Rationale | Duration and Rationale | Frequency |
| | | Postpartum | Postnatal | | | |
| *Guidelines by international organisations* | | | | | | |
| **Europe [23]** European Board & College of Obstetricians and Gynaecologists | **2014** Obstetric and Neonatal Services Standards of Care for Women's Health in Europe | + | + | Not given | Not given | Not given |
| **World Health Organization [3]** | **2022** WHO recommendations on maternal and newborn care for a positive postnatal experience | + | + | **6 weeks** The most critical period following the birth of the newborn for the women, newborns, partners, parents, caregivers, and families. Maternal and neonatal morbidity and mortality are highest during this period | **6 weeks** Same rationale as for the postpartum period | • Within 24 hours after childbirth • On day 3 (48–72 hours) • Between days 7 and 14 • At 6 weeks |
| *Country-level guidelines* | | | | | | |
| **Australia [12]** Council of Australian Governments | **2019** Woman-centered Care Strategic Directions for Australian maternity services | - | + | Not given | **52 weeks (1 year)** No rationale provided | Not given |
| **Canada [9]** Public Health Agency of Canada | **2020** Family-centered Maternity and Newborn Care: National Guidelines | + | + | Not given | **6 weeks** The guideline follows the convention of maternal and newborn care covering 6 weeks postpartum | National uniformity is not present with each Province determining its postpartum care plan |
| **India [8]** National Rural Health Mission | **2021** Induction Training Module for ASHAs | - | + | Not given | **6 weeks** Not given | • For home delivery visits on days:1, 3, 7, 14, 21, 28, & 42 • For institutional delivery visits on days —3, 7, 14, 21, 28, and 42. |
| **South Africa [7] \*\*** Department of Health | **2016** Guidelines for maternity care in South Africa | + | + | Not given | **6 weeks** Not given | • Immediately • 3 to 6 days • 6 weeks |
| **UK [10]** National Institute for Health and Care Excellence (NICE) & The Royal College of Obstetricians and Gynaecologists | **2021** Postnatal care | + | + | Not given | **8 weeks** To cater to the most critical period following birth. | • Within 36 hours after discharge • Health worker home visit within 7–14 days • 6–8 weeks by GP |

*(Continued)*

**Table 1.** (Continued)

| International Body/ Country Institution | Year of publication of the latest guideline/ Position Paper/ Strategy | Use of 'Postpartum' or 'Postnatal' term to refer to care/ services of mothers *after* delivery* | | Postpartum/ postnatal period | Postpartum/ postnatal care provision | |
|---|---|---|---|---|---|---|
| | | Mother | | Duration and Rationale | Duration and Rationale | Frequency |
| | | Postpartum | Postnatal | | | |
| USA [11] *** American College of Obstetricians and Gynaecologists (ACOG) | 2018 (Reaffirmed 2021) Optimizing Postpartum Care | + | - | Not given | 12 weeks The postpartum period is considered the fourth trimester in maternal care and the most challenging for women which includes physical, psychological, social, and sexual health challenges | • Blood pressure check- 3–10 days • High-risk contacts: 1–3 weeks • If necessary: 3–12 weeks • 4–12 weeks- comprehensive postpartum visit |

[a]Only the latest guideline presented from each country/ organisation.

* Use of 'Postnatal' or 'Postpartum' term to refer to care/ services for mothers following childbirth No:—Yes: +

** South African Department of Health published the South African maternal, perinatal, and neonatal health policy in 2021. However, the authors selected a guideline published in 2016 as it details maternal and newborn care in South Africa and Policy (2021) only provides the direction and overarching vision.

***White House of United States published White House blueprint for addressing the maternal health crisis in 2022. However, the authors selected a guideline published in 2018 as it details maternal care provision after delivery. The White House blueprint for addressing the maternal health crisis, 2022 only provides the direction and overarching vision.

documents, which exclusively used the terms postnatal and postpartum, respectively. Interestingly, on at least two occasions (Box 1) WHO highlighted confusion in the use of terms. However, no further explanation has been given for the use of a single term and the ongoing use of the terms interchangeably in the guideline documents included in this review.

The length of the postpartum period was first noted in the document published in 1998, followed by documents from 2010, 2013, and 2022 (Fig 2). Except for the 2013 guideline, where it is indicated as days and weeks following childbirth, all other documents indicated six weeks as the length of the postpartum period. The latest document (2022) roughly defined the

---

### Box 1. Selected phrases from 1998 and 2010 WHO guidelines

WHO (1998) p.7-8

*The words "postpartum" and "postnatal" are sometimes used interchangeably. In this report we use the word "postpartum", except in sections exclusively dealing with the infant. In those sections the word "postnatal" is used.*

WHO (2010) pg. 12

*The terms "postpartum period" and "postnatal period" are often used interchangeably but sometimes separately, when "postpartum" refers to issues pertaining to the mother and "postnatal" refers to those concerning the baby.. . .*

*For care after childbirth, the panel agreed that adopting just a single term would aid clarity. Therefore, the panel agreed that the term "postnatal" should be used for all issues pertaining to the mother and the baby after birth.*

---

| Postpartum | Postnatal | Duration | Title and Year of publication | Service Length | Service Frequency |
|---|---|---|---|---|---|
| | | | **Use of 'Postpartum' or 'Postnatal' term to refer care/ services of mother *after* delivery** | | |
| − | + | Not given | **1976** New Trends and Approaches in the Delivery of Maternal and Child Care in Health Services [17] | Not given | Not given |
| + | − | Not given | **1996** Mother-Baby Package: Implementing safe motherhood in countries [18] | Not given | Not given |
| + | + | 6 weeks | **1998** Postpartum care of the mother and newborn : a practical guide [1] | 6 weeks | • 6 hours (6-12 hrs) • 6 days (3-6 days) • 6 weeks • 6 months |
| + | + | Not given | **2003** Pregnancy, Childbirth, Postpartum and Newborn Care: A guide for essential practice [19] | 6 weeks | • Within 1st week, preferably within 2-3 days • 4-6 weeks |
| + | + | Not given | **2009** WHO Recommended Interventions for Improving Maternal and Newborn Health [20] | 6 weeks | Not given |
| + | + | 6 weeks | **2010** WHO Technical Consultation on Postpartum and Postnatal Care [2] | 6 weeks | • Within one hour • 72 hours • 10–14 days • 6–8 weeks |
| + | + | Not given | **2013** Counselling for maternal and newborn health care: a handbook for building skills [21] | 6 weeks | • Within 24 hours • Within 1 week, preferably on day 3 • 7-14 days • 4-6 weeks |
| + | + | Days and weeks following childbirth | **2013** WHO recommendations on Postnatal care of the mother and newborn [22] | 6 weeks | • Within 24 hours • Day 3 (48-72 hours) • 7–14 days • 6 weeks |
| + | + | Not given | **2015** Pregnancy, Childbirth, Postpartum and Newborn Care: A guide for essential practice [14] | 6 weeks | • Within 24 hours after childbirth. • Day 3 (48-72 hours) • 7-14 days • Clinic visit: 6 weeks |
| + | + | 6 weeks | **2022** WHO recommendations on maternal and newborn care for a positive postnatal experience [3] | 6 weeks | • Within 24 hours • On day 3 (48-72 hours) • 7-14 days • 6 weeks |

**Fig 2. The evolution (1976 to 2022) of routine postpartum care guidelines by the WHO. [1–3, 14, 17–22].**

postpartum period as the most critical period following the birth of the newborn for women, newborns, partners, parents, caregivers, and families. It also referenced that maternal and neonatal morbidity and mortality are highest during this period (Table 1).

While there was no mention of the length and frequency of service provision for routine maternal care after delivery in documents prior to 1998, all documents since have specified the length of service provision during the postpartum period to be six weeks. Additionally, documents post-1998 identified the initiation of the postpartum/ postnatal period to be immediately after delivery, whereas the 1998 document indicated routine postpartum care to commence within an hour post-childbirth.

Interestingly, the 1998 document also indicated the schedule of four contacts with health services at approximately six hours, six days, six weeks, and six months, thereby contradicting its statement of six weeks of routine postpartum/ postnatal service provision with the final contact at six months. The subsequent guideline (2003) discontinued the fourth contact schedule, but this was reintroduced in 2010 with all four contacts to conclude in the first 6 weeks after delivery. While the 2010 guideline recommended first contact to be completed within one hour of childbirth, later guidelines extended the time to up to 24 hours after childbirth. The other three contacts are usually between 48–72 hours, 7–14 days, and six weeks since 2010.

## Global and country-level guidelines: Terminology in relation to the postpartum period and provision of routine care

None of the country-level guidelines mentioned the definition or the length of the postpartum/ postnatal period (Table 1). Similarly, the document from the European EBCOG did not include any relevant information regarding the definition and length of the postpartum/postnatal period. As for WHO, the terms 'postpartum' and postnatal' have been used interchangeably in these documents to refer to care or service provision for mothers following childbirth. Interestingly, in their latest iteration, India (2021) and Australia (2019) have exclusively used the term 'postnatal', while the USA (2018 reaffirmed in 2021) has opted for 'postpartum' (S1 Table).

As for WHO, Canada, India, and South Africa have indicated six weeks for the provision of services for postpartum/ postnatal care. In contrast, Australia, the UK, and the USA (i.e., ACOG) indicate one year (52 weeks), 12 weeks, and eight weeks, respectively, for the provision of routine care.

Similar to the concept of the length of service delivery, the frequency of service delivery i.e., contact points, varies substantially between countries, ranging from two to six contacts at various times. Additional exceptions were Australia, where no detail is provided in the document; and Canada, where each province determines the frequency and the method of contact (Table 1 and S1 Table).

## Discussion

The interchangeable use of the terms 'postpartum' and 'postanal' is common. The only exceptions are the latest online documents from Australia, India, and the USA. While Australia and India have exclusively used the term postnatal, the USA has preferred the term postpartum. Except for WHO, the guideline documents on maternal care after delivery from international and country-level organisations do not define the postpartum period and its duration. Furthermore, the length and schedule of routine postpartum care vary substantially ranging from six weeks to one year, with two to six healthcare contacts, respectively. These inconsistencies in terminology, length, service duration, and schedule could result in confusion when consolidating the evidence required for developing and updating clinical guidelines, and therefore, need harmonisation.

### Postpartum versus postnatal: What's in the term?

International Classification of Diseases Version 11 (ICD-11) exclusively uses the term postpartum for mothers *after* delivery. Some examples include MF35—Postpartum symptom or complaint [24] and QA48—Postpartum care or examination [25], referring to illnesses, ailments, and maternal care post-delivery, respectively. On the other hand, the terms postnatal and perinatal are reserved for the newborn, e.g., KB86—Postnatal intestinal perforation under digestive system disorders of the foetus or newborn [26]. Since the ICD system is used internationally to

compare the causes of morbidity and mortality over time and across countries, it is prudent to follow that system and to reduce the ambiguity of terms in clinical and strategic documents.

Secondly, the term postpartum is more widely used in research and scientific publications. To illustrate: we employed the Web of Science database to estimate which term (postpartum or postnatal) is commonly used in records when referring to mothers *after* delivery published in the year 2023. The test rendered 1,057 original research articles when the search included *((((AB = (woman)) OR AB = (women)) OR AB = (mother*)) AND TI = (postpartum))*. In contrast, only 216 records were identified when *postpartum* was replaced with *postnatal* in the above search criteria. This substantial disparity between the usage of the two terms highlights the partiality of the scientific community towards the use of the term postpartum over postnatal in the context of mother a*fter* delivery.

## Postpartum period and service schedule: Does duration matter?

There is no universal consensus on the duration of the postpartum period. This could be because physiologically, different organs and systems return to the pre-pregnancy stage at varying speeds [5, 27]. If we consider that the duration of routine care for the mother *after* delivery proposed in the strategy guidelines is the default length, then the postpartum period lasts anywhere between six weeks and 52 weeks, as prescribed in the documents reviewed.

Early support of the mother *after* delivery is crucial to limit maternal morbidity and mortality due to infection, haemorrhage, etc., after childbirth [28]. Support in later times is equally critical to ensure the health and well-being of the mother and her offspring. Documents from the WHO and other countries indicate that the six-week visit (often the last health services contact) should emphasise the importance and need of continuous breastfeeding, contraception, and screening for postpartum depression [5, 28].

Proper contraception to support the optimum time between childbirth and the next pregnancy is important for the short and long-term health outcomes of the mother and the newborn [29]. It is recommended that the minimum birth/delivery to the next pregnancy interval should be six months [30–32]. However, in most of the world, formal healthcare contact during the postpartum lasts only for six weeks (as shown in Table 1), thereby limiting the opportunities for providing education to new mothers. A low understanding of the advantages of birth spacing is potentially a predictor for short intervals in addition to low socio-economic and educational status, and a patriarchal society [33, 34].

Optimum birth spacing is critical to ensure the adequate recovery time to maintain overall health following the nutrition and energy burden of the completed pregnancy [35, 36]. Further, it allows the mother to manage their postpartum weight and reduce the risk of entering the subsequent pregnancy at a higher weight status. High preconception weight often leads to high gestational weight gain [37]. Both of these factors are independent predictors of negative short-term outcomes such as gestational diabetes [38], as well as long-term risk of diabetes and cardiovascular disease for the mother [39, 40]. An additional outcome of this weight retention and higher gestational weight gain is the negative impact on mental health [41, 42].

Psychological i.e., emotional, and mental health is equally important as physiological health. Globally, there has been a sharp increase in late maternal deaths i.e., deaths occurring between 43 days and one year after delivery, in middle- and high-income countries when viewed as a proportion against total maternal deaths [43]. Mental health disorders including postpartum depression often lead to discontinuation of breastfeeding, increased risk of baby neglect, family dysfunction, increased medical care and cost, and sometimes suicide. In the USA, late maternal mortality (i.e., death between 43 days and one year after delivery) from mental health conditions quadrupled between 2007 and 2016 [44]. Similar data is available from Australia, with the state of Victora

attributing close to 50% of its total maternal deaths to suicide in the year 2016 with 37.5% of these deaths occurring between 43 days to one year [45]. The phenomenon of suicide in the latter half of the year after delivery is not limited to high-income countries., However, the data is not clear from low- middle-income countries. These deaths are often grouped into the 'comprehensive maternal death' code in ICD-11 due to incomplete reporting and misclassification making it difficult to fully understand the extent of the problem in resource-poor countries [46, 47].

## Where to next?

Considering the information discussed above, we have three suggestions. It would be prudent to use the term postpartum over postnatal when referring to the mother *after* delivery. Secondly, the length of the postpartum period needs to be harmonised and defined in the relevant documents. Finally, the possibility of introducing an additional health service contact for mothers sometime between four to six months after delivery. Interestingly, a six-month contact was included in the WHO documents in 1998 with a focus on general health, contraception, and continuing morbidity at this last visit [1], however, it was not incorporated in the subsequent documents and no explanation was provided on this change. While additional contact(s) between six and 12 months will also be useful, some countries may find it difficult to resource them. Hence, our suggestion of a contact between four and six months *after* delivery will serve as a refresher/reminder on contraception, weight management, and birth spacing [1]. Additionally, this visit could be used for screening and providing support for postpartum depression, and excessive weight retention post delivery, both of which are on the rise globally including low and middle income countries [48, 49].

Further, with the rising prevalence of maternal morbidity and mortality from mental health conditions, Australian documents suggest screening all mothers for postpartum depression between six to 12 weeks after delivery [50]. Similarly, the American Academy of Pediatrics recommends screening at one, two, four, and six months [51]. The UK has refrained from these additional contacts for mental health conditions due to the lack of appropriate screening tools as they tend to provide a high number of false positives [52].

The WHO's key priority of improving maternal health is grounded in efforts for universal health coverage and human rights [53]. WHO generates data, research, and clinical guidelines and supports countries in implementing strategies to promote access to quality health services for ending preventable maternal morbidity and mortality. These strategies and health services are the key pillars for achieving the minimum standard to support the health of a mother and her newborn. While this important work has led to significant reductions in early maternal deaths as a consequence of the common direct causes of maternal injury and death, such as blood loss, infection, etc., the balance is slowly shifting in favour of late maternal deaths due to mental health and lifestyle factors. The rise in late maternal morbidity and mortality has influenced the USA to extend maternal healthcare financial support to 12 months postpartum [54]. Therefore, the increase in late maternal deaths must be considered when planning and implementing further strategies for reducing maternal deaths.

While high-income countries can change and enforce additional strategies and policies for the provision of extended service to mothers, this may not be easy for middle- and low-income countries where a large disparity exists between the rich and poor in receiving healthcare. The disparity in the use of healthcare is also prominent in high-income countries like the USA where women with private healthcare have a higher rate of primary care during the first year after delivery than those without private health coverage [55]. Similar inequalities in the use of postpartum care have been reported in developing countries including South Asia with the poor at a disadvantage [56, 57].

## Strengths and limitations

Using a systematic approach in searching and retrieving the documents from international organisations, G20 and OECD countries allowed for the review of broader global information to access the similarities and differences between countries. However, the inclusion criteria of documents available online and published in English may have led to missing any guidelines that are different from those reported in this review.

## Conclusions

Through this review, we make a case for universal harmonisation of the term 'postpartum' when referring to mothers *after* delivery; add clarity to the strategic documents and guidelines pertaining to the rationale for, and duration of, the postpartum length, and, extend the routine postpartum care schedule. These changes have the potential to support the WHO's recent strategies toward ending preventable maternal mortality [58] and ensure that mothers achieve the best health and well-being outcomes for their future pregnancies and long-term health.

## Supporting information

**S1 File. Search strategy.**
(PDF)

**S1 Table. Summary of findings on routine postpartum care published by international organisations, G20 and OECD countries.**
(PDF)

## Author Contributions

**Conceptualization:** Malith Kumarasinghe, Andrew P. Hills, Kiran D. K. Ahuja.

**Data curation:** Malith Kumarasinghe.

**Formal analysis:** Malith Kumarasinghe, Kiran D. K. Ahuja.

**Methodology:** Malith Kumarasinghe, Manoja P. Herath, Andrew P. Hills, Kiran D. K. Ahuja.

**Project administration:** Malith Kumarasinghe.

**Resources:** Kiran D. K. Ahuja.

**Software:** Kiran D. K. Ahuja.

**Supervision:** Andrew P. Hills, Kiran D. K. Ahuja.

**Validation:** Kiran D. K. Ahuja.

**Visualization:** Kiran D. K. Ahuja.

**Writing – original draft:** Malith Kumarasinghe.

**Writing – review & editing:** Malith Kumarasinghe, Manoja P. Herath, Andrew P. Hills, Kiran D. K. Ahuja.

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
