## [Decision Letter · Decision Letter 0]

9 Jan 2024

PONE-D-23-34077Postpartum versus postnatal period: Do the name and duration matter?PLOS ONE

Dear Dr. Ahuja,

Thank you for submitting your manuscript to PLOS ONE. After careful consideration, we feel that it has merit but does not fully meet PLOS ONE’s publication criteria as it currently stands. Therefore, we invite you to submit a revised version of the manuscript that addresses the points raised during the review process.

The manuscript has been evaluated by two reviewers, and their comments are available below.

The reviewers are positive about your submission, but make several requests for clarification.Could you please carefully revise the manuscript to address all comments raised?

We look forward to receiving your revised manuscript.

Kind regards,

Steve Zimmerman, PhD

Senior Editor, PLOS ONE

Journal Requirements:

Reviewers' comments:

Reviewer's Responses to Questions

**Comments to the Author**

1. Is the manuscript technically sound, and do the data support the conclusions?

Reviewer #1: Yes

Reviewer #2: Yes

2. Has the statistical analysis been performed appropriately and rigorously? 

Reviewer #1: N/A

Reviewer #2: N/A

3. Have the authors made all data underlying the findings in their manuscript fully available?

Reviewer #1: Yes

Reviewer #2: Yes

4. Is the manuscript presented in an intelligible fashion and written in standard English?

Reviewer #1: Yes

Reviewer #2: Yes

5. Review Comments to the Author

Reviewer #1: Thank you for the opportunity to review this paper. The topic is extremely important, and the authors provide a succinct overview of the use of the different terms and the ways to move forward. Below I have minor comments to strengthen the manuscript. Thank you for your work on this topic.

- In your results, you say the initial search yielded 57 documents from 28 countries, but with the additional search, you added one more country-level document. However, in the following statement, you then say the search yield 58 documents from 32 countries. Why is there a discrepancy in countries available from 28 to 32 but you only added one country?

- Please spell out UK and US at first use (line 129)

- Canada has provinces not states (line 193 and table)

Reviewer #2: Although terms postpartum and postnatal are used interchangeably by some as the authors state, in the abstract the focus is on the term postpartum in the results section. Just some suggestions to possibly make this manuscript stronger:

1. Title - Does the....

2. Line 150 - From several recent WHO publications, it looks like they use the term postnatal instead of postpartum pretty consistently. I believe stating this here would provide some additional clarity.

3. Just some additional points about the US for you to consider: Medicaid extension (public insurance) has been recently changed to include the postpartum period up to one year of giving birth and the Pregnancy Mortality Surveillance System (PMSS) defines a pregnancy-related death as a death while pregnant or within 1 year of the end of pregnancy from any cause related to or aggravated by the pregnancy. For maternal deaths in the US - the time frame is extended up to one year. You may want to consider adding documents that focus on capturing maternal deaths as they too might different in how the length of the postpartum period is defined. (https://www.hhs.gov/about/news/2023/12/15/hhs-to-improve-maternal-health-outcomes-with-new-cms-care-model-that-expands-access-to-services-other-proven-maternal-health-approaches.html)

4. Around line 235 - I would suggest extending this paragraph to talk about the timeline used by countries to measure maternal mortality since this time frame also varies from 42 days by WHO to up to one year in the US.

5. Line 278 - how did you decide on 4-6 months here?

6. Line 284 - In the US ACOG suggests a contact with the client within the first three weeks postpartum. (https://www.acog.org/clinical/clinical-guidance/committee-opinion/articles/2018/05/optimizing-postpartum-care)

The authors provided a thorough review of the terms and timing of postpartum care and make the case for the need to choose one term to be used throughout the world.

Thank you for the opportunity to review this manuscript.

6. PLOS authors have the option to publish the peer review history of their article (what does this mean?). If published, this will include your full peer review and any attached files.

Reviewer #1: No

Reviewer #2: No

---

## [Author Response · Author response to Decision Letter 0]

14 Jan 2024

please see the attached document "Response to Reviewers"

---

## [Decision Letter · Decision Letter 1]

18 Feb 2024

PONE-D-23-34077R1Postpartum versus postnatal period: Do the name and duration matter?PLOS ONE

Dear Dr. Ahuja,

Thank you for submitting your manuscript to PLOS ONE. After careful consideration, we feel that it has merit but does not fully meet PLOS ONE’s publication criteria as it currently stands. Therefore, we invite you to submit a revised version of the manuscript that addresses the points raised during the review process.

You have addressed all the comments that were raised by the reviewers. However, there are a few issues to address and your paper will be ready for publication. These issues relate to the time frame the search was conducted and the type of review.

We look forward to receiving your revised manuscript.

Kind regards,

Douglas Aninng Opoku, MPH

Academic Editor

PLOS ONE

Journal Requirements:

Additional Editor Comments:

1. Indicate the type of review

2. Indicate the time frame of the search (date, from when to when)

Reviewers' comments:

Reviewer's Responses to Questions

**Comments to the Author**

1. If the authors have adequately addressed your comments raised in a previous round of review and you feel that this manuscript is now acceptable for publication, you may indicate that here to bypass the “Comments to the Author” section, enter your conflict of interest statement in the “Confidential to Editor” section, and submit your "Accept" recommendation.

Reviewer #1: All comments have been addressed

Reviewer #2: All comments have been addressed

2. Is the manuscript technically sound, and do the data support the conclusions?

Reviewer #1: (No Response)

Reviewer #2: Yes

3. Has the statistical analysis been performed appropriately and rigorously? 

Reviewer #1: (No Response)

Reviewer #2: N/A

4. Have the authors made all data underlying the findings in their manuscript fully available?

Reviewer #1: (No Response)

Reviewer #2: Yes

5. Is the manuscript presented in an intelligible fashion and written in standard English?

Reviewer #1: (No Response)

Reviewer #2: Yes

6. Review Comments to the Author

Reviewer #1: (No Response)

Reviewer #2: The authors have addressed my original comments/questions thoroughly. The use of the term postpartum should be clarified and used consistently by all countries.

7. PLOS authors have the option to publish the peer review history of their article (what does this mean?). If published, this will include your full peer review and any attached files.

Reviewer #1: No

Reviewer #2: No

---

## [Author Response · Author response to Decision Letter 1]

20 Feb 2024

kindly check the Response to Reviewers document. please note - I have tried to change the sequence of files many a times, however the system is not updating it for me. I apologies for this inconvenience.

---

## [Editor Report · Decision Letter 2]

22 Feb 2024

Postpartum versus postnatal period: Do the name and duration matter?

PONE-D-23-34077R2

Dear Dr. Ahuja,

We’re pleased to inform you that your manuscript has been judged scientifically suitable for publication and will be formally accepted for publication once it meets all outstanding technical requirements.

Kind regards,

Douglas Aninng Opoku, MPH

Academic Editor

PLOS ONE

Additional Editor Comments (optional):

Congratulations 